# Comparative Chloroplast Genomics of 21 Species in Zingiberales with Implications for Their Phylogenetic Relationships and Molecular Dating

**DOI:** 10.3390/ijms241915031

**Published:** 2023-10-09

**Authors:** Dong-Mei Li, Hai-Lin Liu, Yan-Gu Pan, Bo Yu, Dan Huang, Gen-Fa Zhu

**Affiliations:** Guangdong Provincial Key Lab of Ornamental Plant Germplasm Innovation and Utilization, Environmental Horticulture Research Institute, Guangdong Academy of Agricultural Sciences, Guangzhou 510640, China; liuhailin@gdaas.cn (H.-L.L.); yangupan163@163.com (Y.-G.P.); yubo@gdaas.cn (B.Y.); huangdan@gdaas.cn (D.H.)

**Keywords:** Zingiberales, chloroplast genome, comparative genomics, phylogenetic relationships, divergence time

## Abstract

Zingiberales includes eight families and more than 2600 species, with many species having important economic and ecological value. However, the backbone phylogenetic relationships of Zingiberales still remain controversial, as demonstrated in previous studies, and molecular dating based on chloroplast genomes has not been comprehensively studied for the whole order. Herein, 22 complete chloroplast genomes from 21 species in Zingiberales were sequenced, assembled, and analyzed. These 22 genomes displayed typical quadripartite structures, which ranged from 161,303 bp to 163,979 bp in length and contained 111–112 different genes. The genome structures, gene contents, simple sequence repeats, long repeats, and codon usage were highly conserved, with slight differences among these genomes. Further comparative analysis of the 111 complete chloroplast genomes of Zingiberales, including 22 newly sequenced ones and the remaining ones from the national center for biotechnology information (NCBI) database, identified three highly divergent regions comprising *ccsA*, *psaC*, and *psaC*-*ndhE*. Maximum likelihood and Bayesian inference phylogenetic analyses based on chloroplast genome sequences found identical topological structures and identified a strongly supported backbone of phylogenetic relationships. Cannaceae was sister to Marantaceae, forming a clade that was collectively sister to the clade of (Costaceae, Zingiberaceae) with strong support (bootstrap (BS) = 100%, and posterior probability (PP) = 0.99–1.0); Heliconiaceae was sister to the clade of (Lowiaceae, Strelitziaceae), then collectively sister to Musaceae with strong support (BS = 94–100%, and PP = 0.93–1.0); the clade of ((Cannaceae, Marantaceae), (Costaceae, Zingiberaceae)) was sister to the clade of (Musaceae, (Heliconiaceae, (Lowiaceae, Strelitziaceae))) with robust support (BS = 100%, and PP = 1.0). The results of divergence time estimation of Zingiberales indicated that the crown node of Zingiberales occurred approximately 85.0 Mya (95% highest posterior density (HPD) = 81.6–89.3 million years ago (Mya)), with major family-level lineages becoming from 46.8 to 80.5 Mya. These findings proved that chloroplast genomes could contribute to the study of phylogenetic relationships and molecular dating in Zingiberales, as well as provide potential molecular markers for further taxonomic and phylogenetic studies of Zingiberales.

## 1. Introduction

Zingiberales consists of eight families, approximately 110 genera, and more than 2600 species [1,2,3,4,5,6]. The eight families in the order are Musaceae, Strelitziaceae, Lowiaceae, Heliconiaceae, Zingiberaceae, Costaceae, Cannaceae, and Marantaceae. The fIRst four families are also called “banana families”, and the last four families are known as “ginger families”. The abundance of the species between these families is imbalanced, with relatively few species found within the Lowiaceae family. The order is morphologically highly diverse and widely distributed in the tropics [7,8]. Many species of the Zingiberales have important economic value as foods, medicines, spices, and ornamentals. Examples include the edible bananas (*Musa balbisiana* Colla and *Musa acuminata* Colla), gingers (*Zingiber officinale* Rosc.), traditional Southern medicines (*Alpinia oxyphylla* Miq. and *Amomum villosum* Lour.), turmeric powder (*Curcuma longa* L.), and ginger lily (*Hedychium coronarium* J. Konig) [9,10,11,12,13,14,15]. Some other species with important ecological value exist as well, such as the Neotropical spIRal gingers (*Costus* L.) [16,17].

A strong and well-resolved phylogeny provides a critical underpinning, not only for systematic studies, but also for investigations of its molecular evolution and comparative genetics [18]. Over the past three decades, growing evidence based on both morphological and/or molecular data has consistently identified well-resolved relationships among four ginger families [1,2,3,4,5,6,7,8,9,10,11,12,13,14,15,16,17,19,20,21,22]. The relationships among the other four banana families have been resolved in many different ways, but never strongly supported [1,2,3,4,5,6,7,8,14,20,21,22]. For example, the placement of Musaceae [1,2,4,8], and in some cases Heliconiaceae [3] or Lowiaceae [23], has to date been controversial. According to one opinion, these three families have previously been placed as sisters to all other families in the Zingiberales; however, their placement has not been strongly and consistently supported in any previous studies [1,2,3,4,8,23]. According to another opinion, based on other previous studies [5,6,20], Musaceae was sister to the clade containing Heliconiaceae, which was in turn sister to Strelitziaceae and Lowiaceae with weak-to-strong support (bootstrap, BS = 69–100%, and posterior probabilities, PP = 0.6–1.0). Meanwhile, the phylogenetic relationships among genera and within the genera of Zingiberales (such as *Cornukaempferia*, *Hedychium*, and *Kaempferia*) using different molecular markers, including few chloroplast DNA fragments and nuclear *internal transcribed spacer* (*ITS*) sequences, are also poorly defined [21,24,25]. Recently, complete chloroplast genomes have been successfully used to resolve phylogenetic relationships in some genera of the Zingiberales, such as *Zingiber* [11], *Amomum* [10], *Alpinia* [13], and *Curcuma* [14]. Unfortunately, there have been no reports on complete chloroplast genomes of *Cornukaempferia*. The phylogenetic relationships among *Cornukaempferia*, *Hedychium*, and *Kaempferia* species in Zingiberales using complete chloroplast genomes are still unknown at present. Therefore, resolving the phylogenetic relationships in these three genera using complete chloroplast genomes is an urgent requirement.

In addition to studying phylogenetic relationships in Zingiberales, researchers have also explored the divergence time of Zingiberales, but most studies have been based on gene fragments. Kress and Specht (2006) [7] constructed the phylogenetic divergence time of Zingiberales based on three genes (*atpB*, *rbcL*, and *18S*) from 36 taxa (including 24 species and 12 outgroups) and five calibration points, indicating that Zingiberales emerged approximately 124 million years ago (Mya), with major family-level lineages becoming established approximately 80–110 Mya. Specht (2006) [22] used chloroplast *trnL*-*F* and *trnK* sequence data for 37 taxa and one outgroup, and estimated the divergence time of Costaceae. The results showed that the initial diversification within Costaceae occurred approximately 65 Mya [22]. Givnish et al. (2018) [6] used draft chloroplast genomes generated for 52 species and estimated the divergence time of Zingiberales using five genes (*atpB*, *psaA*, *psbD*, *rbcL*, and *rps4*). The results indicated that the stem and crown nodes of Zingiberales took place 114 Mya and 83 Mya, respectively [6]. More recently, Fu et al. (2022) [9] used five genes (*ccsA*, *matK*, *ndhF*, *rpoC1*, and *rpoC2*) from 61 chloroplast genomes of Zingiberales and one outgroup to estimate the divergence time, showing that the stem and crown nodes of Zingiberales occurred at 98.57 Mya and 87.59 Mya, respectively [9]. Additionally, Ashokan et al. (2022) [26] used three chloroplast markers (*trnK*/*matK*, *trnL*, and *rps16*) plus one nuclear (*ITS*), three fossil calibration constraints, and one secondary calibration to estimate the divergence time of *Hedychium*. The result revealed that *Hedychium* occurred 10.6 Mya [26]. Most of these studies based on gene fragments generated relatively weakly supported relationships; therefore, more evidence is crucial to further exploring the divergence time and to reconstruct the phylogenetic relationships of Zingiberales.

In this study, we sequenced 22 complete chloroplast genomes of 21 species, including 5 genera (*Cornukaempferia*, *Hedychium*, *Kaempferia*, *Calathea* and *Stromanthe*) from two families Zingiberaceae and Marantaceae, and explored the chloroplast genomes structural features and sequences differentiation among these species. Furthermore, we combined them with published chloroplast genomes from the NCBI database, representing eight families of Zingiberales and four outgroups, for reconstructing phylogenetic relationships and estimating the divergence time of Zingiberales. The main aims of this study were to: (1) analyze the structures and features of the newly sequenced complete chloroplast genomes, (2) identify highly variable regions in complete chloroplast genomes as potential DNA markers for future species identification in Zingiberales, (3) infer the molecular evolution of complete chloroplast genomes in Zingiberales, (4) reconstruct the phylogenetic relationships of Zingiberales by chloroplast genomes, especially determining the phylogenetic positions of *Cornukaempferia*, *Hedychium*, and *Kaempferia*, and (5) estimate the divergence time of Zingiberales by chloroplast genomes.

## 2. Results

### 2.1. Characteristics of 22 Complete Chloroplast Genomes in Zingiberales

The 22 newly sequenced samples of Zingiberales generated approximately 222.60 Gb of paired-end clean data, ranging from 5.84 to 13.26 Gb clean data for each sample after removing adapters and low-quality data (Appendix A). All 22 sequenced chloroplast genomes displayed a typical quadripartite structure containing one large single-copy (LSC), one small single-copy (SSC) and two inverted repeat regions (IRa and IRb) by OGDRAW [27] and CGView tool [28] (Figure 1 and Table 1). The 22 complete chloroplast genomes generated here were deposited in GenBank with accession numbers OP805573 to OP805594 (Table 1). Their size ranged from 161,303 bp (*Stromanthe sanguinea*) to 163,979 bp (*Hedychium menghaiense*) (Table 1 and Appendix A). They showed four junction regions, including a separate LSC region of 87,056–91,910 bp, an SSC region of 15,640–20,848 bp, and a pair of IRs (IRa and IRb) of 27,074–29,788 bp each (Figure 1, Table 1 and Appendix A). The GC content of these 22 complete chloroplast genomes was very similar (36.08–36.67%) (Table 1). Specifically, the GC content in the IR regions (41.14–42.16%) was higher than that in the LSC regions (33.84–34.44%) and SSC regions (29.49–30.35%) (Table 1). The GC content of protein-coding genes of these 22 complete chloroplast genomes ranged from 36.91% to 37.60% (Table 1).

In this study, each sequenced chloroplast genome contained 131–134 predicted functional genes, which consisted of 87–88 protein-coding genes, 36–38 tRNA genes, and 8 rRNA genes (Table 1 and Appendix A). Among these genes, a total of 111–112 different genes were detected in these 22 chloroplast genomes, including 79 protein-coding genes, 28–29 tRNA genes and 4 rRNA genes (Table 1 and Appendix A). Although most of the protein-coding genes tRNAs and rRNAs among these 22 chloroplast genomes were similar, slight differences existed, for examples, *rpl22* had two copies only in the chloroplast genome of *Calathea makoyana*. In addition, *trnH-GUG* had only one copy in the chloroplast genome of *Hedychium brevicaule* in which *trnR-UCU* was missing (Table 2 and Appendix A).

In total, 18 genes contained introns in each sequenced chloroplast genome (Table 2). Sixteen genes (*rpoC1*, *rpl2*, *rpl16*, *rps12*, *rps16*, *petB*, *petD*, *atpF*, *ndhA*, *ndhB*, *trnA-UGC*, *trnG-UCC*, *trnI-GAU*, *trnK-UUU*, *trnL-UAA*, and *trnV-UAC*) contained one intron, while *clpP* and *ycf3* each contained two introns (Table 2 and Appendix A). Among the 18 intron-containing genes in these 22 chloroplast genomes, four genes (*ndhB*, *rpl2*, *trnA-UGC* and *trnI-GAU*) occurred in both IRs, 12 genes (*atpF*, *clpP*, *petB*, *petD*, *rpl16*, *rpoC1*, *rps16*, *trnG-UCC*, *trnL-UAA*, *trnK-UUU*, *trnV-UAC* and *ycf3*) were distributed in the LSC, one gene (*ndhA*) was in the SSC, and one gene’s (*rps12*) fIRst exon was located in the LSC with the other two exons in both IRs (Figure 1 and Appendix A).

### 2.2. Repeat Sequences Analyses

In this study, four types of long repeats including forward repeats, palindromic repeats, reverse repeats, and complement repeats were identified in our newly assembled 22 chloroplast genomes (Figure 2A and Appendix A). Among these 22 chloroplast genomes, *C. makoyana* had the largest number of long repeats (384) and *Kaempferia parviflora* had the smallest number of long repeats (51) (Figure 2A and Appendix A). The number of forward repeats varied between 19 (*Cornukaempferia aurantiflora*) and 306 (*C. makoyana*), the number of palindromic repeats varied from 23 (*K. parviflora*) to 70 (*C. makoyana*), the number of reverse repeats varied between 3 (*K. parviflora*) and 21 (*Hedychium villosum* var. *tenuiflorum*), and the number of complement repeats varied from 1 (*C. makoyana* and *Hedychium yunnanense*) to 9 (*H. villosum* var. *tenuiflorum*) (Figure 2A and Appendix A). Long repeats with 30–34 bp were found to be the most common in these 22 chloroplast genomes (Figure 2B and Appendix A). Long repeats with lengths of ≥45 bp and lengths of 35–39 bp were the second and third most common, respectively (Appendix A).

Six types of simple sequence repeats (SSRs) were discovered among these 22 chloroplast genomes: mononucleotide, dinucleotide, trinucleotide, tetranucleotide, pentanucleotide and hexanucleotide (Figure 3A and Appendix A). In total, there were 85 to 127 SSRs in each chloroplast genome (Figure 3A). Among these SSRs, only the chloroplast genome of *C. aurantiflora* had no pentanucleotide repeats, and four chloroplast genomes of *Hedychium chrysoleucum*, *Hedychium tienlinense*, *H. menghaiense*, and *S. sanguinea* had no hexanucleotide repeats (Figure 3A and Appendix A). Among each sequenced chloroplast genome, mononucleotide repeats were the most frequent, with numbers ranging from 41 to 63, followed by dinucleotide repeats, ranging from 14 to 35, tetranucleotide repeats, ranging from 10 to 21, trinucleotide repeats, ranging from 3 to 14, hexanucleotide repeats, ranging from 0 to 7, and pentanucleotide repeats, ranging from 0 to 6 (Figure 3A and Appendix A). The majority of the mononucleotide SSRs were A/T repeats, which accounted for 92.68–100% of all the mononucleotide types among these 22 genomes (Figure 3B and Appendix A). In the dinucleotide repeats, the AG/CT repeats were observed most frequently except in the chloroplast genome of *C. makoyana* (Figure 3B and Appendix A). In the chloroplast genome of *C. makoyana*, AT/AT repeats were the most frequent dinucleotide repeats (Figure 3B and Appendix A). In the tetranucleotide category, the AAAT/ATTT repeats were the most abundant type (Figure 3B and Appendix A). SSRs were more frequently located in the LSC regions (54–87 loci, 62.09–73.68%) than in the SSC regions (15–23 loci, 14.02–22.35%), IRa regions (3–10 loci, 3.53–8.40%), and IRb regions (3–9 loci, 3.53–8.05%) among these 22 chloroplast genomes (Appendix A).

### 2.3. Codons Usage Analysis

The 22 newly sequenced chloroplast genomes of Zingiberales were analyzed to survey information on the codon usage, amino acid frequency, and relative synonymous codon usage (RSCU) (Appendix A). The total codons (excluding stop codons) of these 22 chloroplast genomes ranged from 25,908 (*S. sanguinea*) to 27,694 (*H. brevicaule*). Of these, 61 codons encoded 20 amino acids (Figure 4 and Appendix A). The codons ATG and TGG, encoding methionine (Met) and tryptophan (Trp), respectively, showed no codon bias, both with RSCU values of 1.00 among these 22 chloroplast genomes (Figure 4 and Appendix A). The codons of four with the highest RSCU values (AGA, TTA, GCT, and TCT) and six with the lowest RSCU values (AGC, GGC, CGC, GCG, CTG and GAC) were found in these 22 chloroplast genomes (Figure 4 and Appendix A). Twenty-nine codons showed codon usage bias with RSCU > 1.00 in protein-coding genes of 21 chloroplast genomes; however, in protein-coding genes of *C. makoyana* chloroplast genome, thirty codons indicated codon usage bias with RSCU > 1.00 (Appendix A).

### 2.4. Comparative Analysis of 22 Complete Chloroplast Genomes in Zingiberales

Using the complete chloroplast genome of *H. bijiangense* as the reference, 22 newly sequenced chloroplast genomes of Zingiberales were compared by using mVISTA [29] and CGView [28] (Figure 1 and Figure 5). The mVISTA result revealed that the two IR regions were less divergent than the LSC and SSC regions (Figure 5). The non-coding regions showed obviously higher divergence than the protein-coding regions (Figure 5). The main divergences for the protein-coding regions were *trnQ*, *rpl33*, *trnG*, *ycf3*, *trnS* and *ccsA*. For the non-coding regions, highly divergent regions were *accD-psaI*, *rpl32-trnL*, *rbcL-accD*, and *ndhG-psaC* (Figure 5). The CGView result also indicated that the LSC and SSC regions were significantly more divergent than the two IR regions, and the main divergences originated from LSC and SSC regions (the innermost fourth color ring to the outwards 25th ring in Figure 1). Compared to the chloroplast genome of *H. bijiangense* (the innermost fourth color ring in Figure 1), the other 21 complete chloroplast genomes showed four divergent regions in LSC (*trnG-trnS*, *rbcL-accD*, *psaA-ycf3* and *ycf4-cemA*), one region in SSC (*ndhG-psaC*) and one region in IRa (*ycf1*).

The borders of LSC/IRb, IRb/SSC, SSC/IRa, and IRa/LSC among these 22 chloroplast genomes were compared and shown in detail (Figure 6). Among these 22 chloroplast genomes, the *rps19* and *psbA* genes were located in the IRa/LSC borders, respectively (Figure 6). The distances between the ends of *rps19* and the IRa/LSC borders were 120–305 bp, and the distances between the start of *psbA* and the IRa/LSC boundaries ranged from 93 bp to 291 bp (Figure 6). For the LSC/IRb borders, the *rpl22* and *rps19* genes were located in the LSC/IRb borders in all the 22 chloroplast genomes. A total of 24–95 bp were found between the ends of *rpl22* and the LSC/IRb borders among these 22 chloroplast genomes, and the distances between the start of *rps19* and the LSC/IRb borders ranged from 124 bp to 306 bp (Figure 6). The SSC/IRa borders were located in the *ycf1* gene, which crossed into the IRa region in the 21 chloroplast genomes; however, the lengths of *ycf1* in the IRa regions significantly varied among these 21 chloroplast genomes, ranging from 1050 bp to 3877 bp (Figure 6). In *S. sanguinea*, *ndhF* was found in the SSC/IRa border in its chloroplast genome; and the distance between the end of *ndhF* and the SSC/IRa border was 5 bp (Figure 6). Regarding the IRb/SSC borders, the *ycf1* and *ndhF* genes were located in the IRb/SSC borders in the 21 chloroplast genomes, except in the chloroplast genome of *S. sanguinea*. The *ycf1* gene expanding into the SSC regions ranged from 6 bp to 132 bp, and the distances between the start of *ndhF* and the IRb/SSC borders ranged from 4 bp to 447 bp among these 21 chloroplast genomes (Figure 6). Regarding *S. sanguinea*, only *ycf1* gene was located at the border of IRb/SSC in its chloroplast genome, and the *ycf1* gene expanded into the SSC region by 4107 bp (Figure 6). In conclusion, the IR/LSC borders of these 22 chloroplast genomes of Zingiberales were relatively conserved and similar, but the IR/SSC borders exhibited variations.

### 2.5. Highly Divergence Regions and Selective Pressure Analyses of Zingiberales

Nucleotide diversity (Pi) and single-nucleotide substitutions in the LSC, SSC, IRa, IRb and the total of chloroplast genomes were analyzed among the 111 complete chloroplast genomes of Zingiberales (Figure 7 and Table 3). One hundred and eleven chloroplast genomes of Zingiberales were aligned with a matrix of 163,883 bp with 111,120 variable sites (67.80%) and 92,746 parsimony informative sites (56.59%). The Pi value of the complete chloroplast genomes was 0.1565 (Table 3). The LSC region had the highest Pi value (0.1751) and the IRa region had the lowest Pi value (0.1455) (Table 3). Among the protein-coding regions, the Pi values ranged from 0 to 0.2695 and had an average value of 0.1399 (Appendix A). Four protein-coding regions (*trnM*, *trnL*, *ccsA* and *psaC*) showed remarkably high values (Pi > 0.20; Figure 7A and Appendix A). For the non-coding regions, Pi values ranged from 0 to 0.2379 (*ccsA-ndhD*) and had an average of 0.1423 (Appendix A). Five of these regions had remarkably high values (Pi > 0.20), including *trnQ*-*psbK*, *psbL*-*psbF*, *ccsA-ndhD*, *ndhD*-*psaC*, and *psaC*-*ndhE* (Figure 7B and Appendix A). Considering the results of Pi values and the length of regions ≥ 150 bp for the selection of potential molecular markers of Zingiberales, three regions were identified, including *ccsA*, *psaC* and *psaC*-*ndhE*.

The ratio (ω) of non-synonymous (dN) to synonymous (dS) substitution (dN/dS) for 66 shared protein-coding genes was analyzed across 111 complete chloroplast genomes of Zingiberales. In this study, using the M8 model (β & ω > 1) for estimating gene selection pressure, 32 genes were under positive selection with a posterior probability greater than 0.95 using the BEB method (Table 4). These genes with positive selection sites could be divided into six categories: subunits of photosystem (*psaA*, *psaI*, *psbA*, *psbB* and *psbD*), subunits of cytochrome (*petB* and *petD*), subunits of ATP synthase (*atpA*, *atpB*, *atpF* and *atpI*), subunits of NADH dehydrogenasee (*ndhA*, *ndhB*, *ndhC*, *ndhD*, *ndhF*, and *ndhI*), subunits of ribosome (*rpl20*, *rpl23*, *rps3*, *rps4*, *rps7*, *rps8* and *rps15*) and others (*rpoB*, *rpoC1*, *rpoC2*, *rbcL*, *clpP*, *matK*, *ycf3* and *ycf4*). Among these 32 genes, *clpP* harbored the highest number of positive amino acids sites (15), followed by *rbcL* (8), *rpoC2* (8), *matK* (7), *ndhB* (5), *rps7* (5), *atpB* (4), *atpA* (3), *atpF* (3), *ndhF* (3), *psaA* (3), *psbD* (3), and *rpl20* (3); the remaining 19 genes had one or two positive amino acids sites (Table 4).

### 2.6. Phylogenetic Relationships of Zingiberales

In this study, we defined strong support as 85% ≤ maximum likelihood bootstrap (MLBS) ≤ 100% and 0.90 ≤ Bayesian inference posterior probabilities (BIPP) ≤ 1.0; moderate support as 70% ≤ MLBS < 85% and 0.80 ≤ BIPP < 0.90; and weak support as MLBS < 70% and BIPP < 0.80. Both ML and BI phylogenetic trees based on chloroplast genomes generated almost identical topological structures with strong support among eight families of Zingiberales (MLBS = 94–100%, and BIPP = 0.93–1.0) (Figure 8). Cannaceae was sister to Marantaceae, forming a clade that was collectively sister to the clade of (Costaceae, Zingiberaceae) with strong support (MLBS = 100%, and BIPP = 0.99–1.0) (Figure 8). Heliconiaceae was sister to the clade of (Lowiaceae, Strelitziaceae), then collectively sister to Musaceae with strong support (MLBS = 94–100%, and BIPP = 0.93–1.0) (Figure 8). The clade of ((Cannaceae, Marantaceae), (Costaceae, Zingiberaceae)) was sister to the clade of (Musaceae, (Heliconiaceae, (Lowiaceae, Strelitziaceae))) with robust support (MLBS = 100%, and BIPP = 1.0) (Figure 8).

Within Marantaceae, *S. sanguinea* was sister to *Phrynium rheedei* with strong support, then forming a clade that was sister to *C. makoyana* with strong support (MLBS = 100%, and BIPP = 1.0). Within Zingiberaceae, *Hedychium* was sister to the clade of (*Monolophus*, *Roscoea*) with strong support (MLBS = 96%, and BIPP = 1.0) (Figure 8). *Kaempferia* was sister to *Zingiber* with strong support (MLBS = 97%, and BIPP = 1.0), then forming a clade that was sister to the clade of *Boesenbergia kingii* and *Curcuma flaviflora* with strong support (MLBS = 99%, and BIPP = 1.0) (Figure 8). *Cornukaempferia* was sister to the clade of ((*B. kingii, C. flaviflora*), (*Kaempferia*, *Zingiber*)) with strong support (MLBS = 86%, and BIPP = 1.0) (Figure 8). Within genus *Hedychium*, there were two unidentified species (*Hedychium* sp.1 LDM232 and *Hedychium* sp.2 LDM222). *Hedychium* sp.1 LDM232 was fIRstly clustered with *H. kwangsiense*, then forming a clade that was sister to *H. brevicaule*, and *Hedychium* sp.2 LDM222 was sister to these three species with moderate to strong support (MLBS = 83–87%, and BIPP = 1.0) (Figure 8).

### 2.7. Divergence Time Estimation of Zingiberales

Divergence time estimation suggested that the stem and crown nodes of Zingiberales were 100.1 Mya (95% HPD: 92.2–109.6 Mya) and 85.0 Mya (95% HPD: 81.6–89.3 Mya), respectively (Figure 9). The crown nodes of ((Cannaceae, Marantaceae), (Costaceae, Zingiberaceae)), (Cannaceae, Marantaceae) and (Costaceae, Zingiberaceae) were 80.5 Mya (95% HPD: 75.4–85.8 Mya), 62.5 Mya (95% HPD: 39.0–76.2 Mya), and 75.2 Mya (95% HPD: 69.8–81.2 Mya), respectively (Figure 9). The crown nodes of Marantaceae, Costaceae and Zingiberaceae occurred at 38.6 Mya (95% HPD: 24.7–58.3 Mya), 34.6 Mya (95% HPD: 22.3–45.4 Mya), and 65.1 Mya (95% HPD: 62.9–69.5 Mya), respectively (Figure 9). Within Zingiberaceae, the species in three genera *Hedychium*, *Kaempferia* and *Cornukaempferia* occurred approximately 8.1 Mya (95% HPD: 4.2–15.4 Mya), 23.5 Mya (95% HPD: 14.8–31.6 Mya), and 34.9 Mya (95% HPD: 28.7–39.9 Mya), respectively (Figure 9).

The crown nodes of (Musaceae, (Heliconiaceae, (Lowiaceae, Strelitziaceae))), (Heliconiaceae, (Lowiaceae, Strelitziaceae)) and (Lowiaceae, Strelitziaceae) were 79.4 Mya (95% HPD: 69.4–86.5 Mya), 71.4 Mya (95% HPD: 55.0–82.4 Mya), and 46.8 Mya (95% HPD: 30.5–70.9 Mya), respectively (Figure 9). The crown nodes of Musaceae, Heliconiaceae, Strelitziaceae and Lowiaceae occurred at 54.9 Mya (95% HPD: 44.9–70.7 Mya), 15.2 Mya (95% HPD: 3.8–36.4 Mya), 20.4 Mya (95% HPD: 7.7–38.1 Mya), and 11.4 Mya (95% HPD: 3.4–26.3 Mya), respectively (Figure 9). Within Musaceae, the crown nodes of *Musa* and *Musella*-*Ensete* clades were 41.2 Mya (95% HPD: 29.4–61.2 Mya), and 41.9 Mya (95% HPD: 40.6–44.8 Mya), respectively (Figure 9).

## 3. Discussion

### 3.1. Genome Structure Comparison and Sequence Variation

In this study, 22 complete chloroplast genomes of Zingiberales showed a typical quadripartite structure (a single LSC region, a single SSC region and a pair of IR regions), which had been reported in other Zingiberaceae and Musaceae species [9,10,11,13,14]. Although most of the protein-coding genes tRNAs and rRNAs were highly conserved, gene loss and gene duplication occurred in these 22 complete chloroplast genomes. For examples, *H. brevicaule* lost *trnR-UCU* and had only one copy of *trnH-GUG* in its chloroplast genome and *C. makoyana* had two copies of *rpl22* only in its chloroplast genome, suggesting that gene loss and insertion had happened during the evolutionary processes of *H. brevicaule* and *C. makoyana*. In contrast, there had been reports of gene loss and duplication in other chloroplast genomes of higher plants, such as losses of *ndh* genes in families Orobanchaceae [30] and Orchidaceae [31], and duplication of *trnS-GCU* and *trnT-UGU* in *Globba schomburgkii* [19]. IR contraction and expansion of chloroplast genomes were recognized to be important evolutionary events and may cause chloroplast genome size variations, production of pseudogenes, gene duplication or the reduction in duplicate genes to single genes [19,30,31]. In this study, the IR region of *C. makoyana* was 27,074 bp long, which was shorter than that of the other 21 chloroplast genomes (Table 1), indicating the other Zingiberales species differentiated later than *C. makoyana*. Additionally, among studied 22 chloroplast genomes herein, only *ycf1* and *ndhF* were found at the IRb/SSC and SSC/IRa borders in the chloroplast genome of *S. sanguinea*, respectively. Therefore, changes in the SSC/IR borders may be the main reasons for the IR contraction and expansion in these Zingiberales species.

Comparative analysis of the 111 complete chloroplast genomes of Zingiberales revealed that the LSC and SSC regions were more divergent than the IR regions (Table 3 and Figure 7), consistent with findings for other plants [12,13,14,15,31]. Previous studies used three combined molecular markers (*atpB*, *rbcL* and *18S*) to confirm the relationships among the ginger families, but were not able to resolve the earliest divergent lineages in Zingiberales [2]. Based on the Pi values studied in the present study, it was also obvious that the frequently used chloroplast genome markers, including *atpB* and *rbcL*, presented relatively low polymorphisms (0.17, 0.16, respectively) at the order level. Based on Pi values studied currently, 3 divergent hotspot regions among 111 complete chloroplast genomes of Zingiberales were identified, including *ccsA*, *psaC* and *psaC*-*ndhE*. By comparison, *ccsA* was also reported as a divergent hotspot region in Myrtales [32] and Monsteroideae [33]. Therefore, these variable regions may be suitable for potential DNA markers for phylogenetic relationships and species identification studies of Zingiberales.

### 3.2. Positive Selection and Phylogenetic Analysis within Zingiberales

In this study, 32 genes with positive selection sites were identified among 111 complete chloroplast genomes of Zingiberales (Table 4). In contrast, we found fewer genes under positive selection than results in a previous study [34], in which most protein-coding genes (62) underwent positive selection pressures among 14 Araceae species. Our results found that 15 positively selected sites were identified in *clpP* genes for Zingiberales, suggesting that *clpP* may play an important role in the adaptive evolution of Zingiberales. Additionally, we found that *rpoC2*, *rbcL* and *matK* also possessed relatively high positive selection sites (8, 8, 7, respectively). Current studies have revealed that these four genes with positive selection in land plants may be very common [31,35,36,37,38]. For examples, *rpoC2* and *rbcL* have been reported as positive selection in orchid species [31]; *matK* and *rbcL* have been identified under positive selection in Ulmaceae species [35]; *rpoC2* and *clpP* have been identified under positive selection in *Anisodus* species [36]; *matK* has been identified under positive selection in the *Cotinus* species [37], and *clpP* has been identified under positive selection in the *Astragalus* species [38]. The high positive selection sites of *clpP*, *rpoC2*, *rbcL* and *matK* indicated that these four genes were valuable markers for the adaptive evolution studies of Zingiberales. On the one hand, Zingiberales species commonly maintained high levels of plant diversity, such as diverse pseudostem heights and leaf sizes; for instance, *Musa coccinea* had pseudostem height of 100–200 cm and leaf size of 50–100 × 10–25 cm, respectively, while *Orchidantha chinensis* and *Roscoea humeana* had pseudostem heights of 50–100 cm and 13–25 cm, respectively, and leaf sizes of 22–30 × 5.5–11 cm and 10–30 × 3–6 cm, respectively [39]. On the other hand, Zingiberales species also owned diverse habitats; for example, *R. humeana* lived in pine forests, meadows and rocky hillside at altitudes of 2900–3800 m, whereas *M. coccinea* scattered in the valley and slope below altitudes of 600 m [39,40]. Therefore, genes of chloroplast genome involved in energy (photosystem, ATP synthase, NADH dehydrogease) and development (ribosome and others), may play important roles during the evolution and adaptation of Zingiberales species to their respective habitats.

Compared with previous studies based on morphology, chloroplast genome genes and nuclear *ITS* [1,3,23], incomplete chloroplast genomes and nuclear genes [4,5,6,20], our results sampled more species using chloroplast genome sequences and showed high resolution of phylogenetic relationships in Zingiberales (Figure 8). Eight clades representing eight families were fully resolved with strong support (Figure 8). Across these previous studies, the four ginger families (Cannaceae, Marantaceae, Costaceae, and Zingiberaceae) and all the relationships among them were strongly supported; however, the placements of the other four banana families (Musaceae, Strelitziaceae, Lowiaceae, and Heliconiaceae) have to date been uncertain and inconsistent. On the one hand, in some previous studies, Musaceae [1,2,4,8] or Heliconiaceae [3] or Lowiaceae [27] were placed as the sister to all the other families in the Zingiberales. On the other hand, Musaceae was sister to the clade containing Heliconiaceae, which was in turn sister to Strelitziaceae and Lowiaceae with weak to strong support (MLBS = 69–100%, and BIPP = 0.6–1.0) [5,6,20]. Our phylogenomic analysis of Zingiberales identified strong support for the sister relationship between Musaceae and a clade of Heliconiaceae + Strelitziaceae-Lowiaceae (MLBS = 94–100%, and BIPP = 0.93–1.0; Figure 8), which was in agreement with the backbone from some previous studies [5,6,20]. Additionally, the phylogenetic positions of *Cornukaempferia*, *Hedychium* and *Kaempferia* within Zingiberaceae were also determined. The present phylogenetic results provided high degree of credibility that complete chloroplast genomes may be useful for phylogeny of Zingiberales in the future. Further study should sample more taxa of Zingiberales and obtain more complete chloroplast genomes data to identify whether our results are in agreement with those from nuclear genes.

### 3.3. Divergence Time within Zingiberales

In this study, the results of divergence time estimation showed that the crown node of Zingiberales most likely occurred at 85.0 Mya (95% HPD: 81.6–89.3 Mya) (Figure 9). The divergence time of Zingiberales estimated herein was in close proximity to some previous reports, such as 87.59 Mya reported by Fu et al. (2022) [9], and 83 Mya reported by Givnish et al. (2018) [6]. However, Kress and Specht (2006) using three genes (*atpB*, *rbcL*, and *18S*) from 36 taxa (including 24 species of Zingiberales and 12 outgroups) and five calibration points, estimated the divergence time of Zingiberales to be approximately 124 Mya [7]. Present analyses estimated that the major family-level lineages of Zingiberales became established approximately 46.8–80.5 Mya (Figure 9), which were younger than the ages estimated by Kress and Specht (2006) (80–110 Mya) [7]. In addition, within Zingiberaceae, the crown node of *Hedychium* had occurred approximately 8.1 Mya (95% HPD: 4.2–15.4 Mya) (Figure 9), which coincided with a report that *Hedychium* was a very young lineage that originated approximately 10.6 Mya [26]. Because taxon sampling, analysis methods, molecular data, and calibration points were different among these divergence time studies [6,7,9,26], we cannot perfectly compare across all these studies. Changes in taxon sampling, analysis methods, molecular data, and calibration points may lead to differences in divergence times estimation in these studies [6,7,9,26].

## 4. Materials and Methods

### 4.1. Plant Sample Collection, DNA Extraction, and Sequencing

Fresh leaves of 22 individuals from 21 species, representing 5 genera and 2 families in Zingiberales, were collected from the resource garden (23°23′ N, 113°26′ E) of the environmental horticulture research institute at the Guangdong Academy of Agricultural Sciences, Guangzhou, China. Each leaf sample was immediately frozen in liquid nitrogen and stored at −80 °C until use. Total chloroplast genomic DNA was extracted from each leaf sample using sucrose gradient centrifugation method with minor modifications [41]. The chloroplast DNA quality and concentration were detected by using 1% (*w*/*v*) agarose gel electrophoresis and NanoDrop 2000 microspectrometer (Wilmington, DE, USA). Each qualified chloroplast DNA (1.0 μg) was used for construction a DNA library with fragments of approximately 350 bp, volume of 25 μL and concentration of 21.5 ng/μL, and then sequenced on an Illumina NovaSeq 6000 platform with 150 bp paired-end reads length (Biozeron, Shanghai, China). The original raw data were filtered by Trimmomatic v. 0.39 with default parameters [42] to delete adaptors and low-quality reads.

### 4.2. Chloroplast Genome Assembly and Annotation

Filtered high-quality clean reads were assembled into complete chloroplast genomes using GetOrganelle v. 1.7.6.1 [43] with default settings. Geneious Prime 2022 (Biomatters Ltd., Auckland, New Zealand) [44] was used for sequence correction with a reference chloroplast genome of *Hedychium coronarium* from Guangdong (GenBank MK262736). Each assembled complete chloroplast genome was annotated using GeSeq [45] and the online Dual Organellar Genome Annotator (DOGMA) [46] with default parameters, respectively. The transfer RNA (tRNA) and ribosomal RNA (rRNA) sequences were confirmed by tRNAscanSE v. 2.0.5 [47] and BLAST v. 2.13.0 [48]. The newly annotated complete chloroplast genome sequences were fIRst validated using online GB2sequin [49], further were verified and formatted using Sequin v. 15.50 from NCBI, and deposited in GenBank (accession numbers are shown in Table 1). Newly complete chloroplast genomes maps were drawn using Organellar Genome Draw (OGDRAW) v. 1.3.1 [27].

### 4.3. Repeat Sequences Analysis

Four types of long repeats of the newly sequenced 22 chloroplast genomes of Zingiberales, namely, forward, palindrome, reverse and complement repeats, were identified using REPuter [50] with a minimal repeat size of 30 bp, a hamming distance of 3 and a repeat identity of more than 90%. Simple sequence repeats (SSRs) of these 22 chloroplast genomes were examined by MISA-web [51] with the following parameters: minimum repeat units of 10 for mononucleotides; 5 for dinucleotides, 4 for trinucleotides, and 3 for tetra-, penta- and hexanucleotides.

### 4.4. Codon Usage Analysis

The relative synonymous codon usage (RSCU) and amino acid frequencies of the 22 chloroplast genomes of Zingiberales were analyzed using MEGA v. 7.0 [52] with default parameters. A RSCU value of >1 indicates that the codon is used more frequently, a RSCU value = 1 indicates that the codon has no use preference, and a value of <1 indicates that the codon is used less frequently. Clustered heat map of RSCU values of newly sequenced 22 chloroplast genomes of Zingiberales was constructed with R v. 4.0.2 (https://www.R-project.org) (accessed on 10 September 2022).

### 4.5. Chloroplast Genomes Comparison

Chloroplast genomes structures among the 22 newly sequenced genomes of Zingiberales were compared with the mVISTA program in the Shuffle-LAGAN mode [29], using the chloroplast genome of *Hedychium bijiangense* as the reference. The newly generated 22 chloroplast genomes of Zingiberales for LSC/IR and SSC/IR boundaries and their adjacent genes were analyzed using IRscope [53]. Furthermore, chloroplast genomes comparisons across 22 chloroplast genomes of Zingiberales were performed using CGView Server [28]. GC contents were detected based on GC skew using the equation: GC skew = (G − C)/(G + C).

### 4.6. Nucleotide Diversity and Gene Selective Pressure Analyses

Nucleotide diversity (Pi) was calculated by sliding window analysis using DnaSP v. 6.12.03 [54]. In total, 111 complete chloroplast genomes of Zingiberales were analyzed, including 22 newly sequenced complete chloroplast genomes and 89 complete chloroplast genomes from the NCBI database (Appendix A). Additionally, variable and parsimony informative base sites of the LSC, SSC, IRa, IRb, and complete chloroplast genomes were also calculated.

To detect positively selected amino acid sites in 111 complete chloroplast genomes of Zingiberales, the nonsynonymous (dN) and synonymous (dS) substitution rates of consensus protein-coding genes were calculated by using the CodeML program implemented in EasyCodeML [55]. Gene selective pressure analysis was based on 66 consensus protein-coding genes sequences after removing all stop codons. The positive selection model of M8 (β & ω > 1) was used to detect positively selected sites based on both the dN and dS ratios (ω) and likelihood ratio tests (LRTs) values [56]. The bayes empirical bayes (BEB) method was used to identify the most likely codons under positive selection, with a posterior probability higher than 0.95 and 0.99 indicating sites under positive selection and strong positive selection, respectively [57].

### 4.7. Phylogenetic Relationship Analysis

To reconstruct and confirm the phylogenetic relationships of eight families in Zingiberales, 115 chloroplast genomes of Zingiberales, including 22 complete chloroplast genomes generated in the present study, 89 complete chloroplast genomes and 4 incomplete chloroplast genomes downloaded from the NCBI GenBank database, were analyzed (Appendix A). *Commelina communis* (MW617984), *Pollia japonica* (MW617990), *Rhopalephora scaberrima* (MW617991), and *Siderasis fuscata* (MW617992) were used as outgroups. Chloroplast genome sequences were aligned using MAFFT v. 7.458 [58] with default parameters, and manually checked when necessary. Phylogenetic tree was constructed using ML and BI methods, respectively. The best nucleotide substitution model (GTR + G + I) was determined using Akaike Information Criterion (AIC) in jModelTest v. 2.1.10 [59]. ML analysis was conducted in PhyML v. 3.0 [60] with 1000 bootstrap replicates. BI analysis was performed in MrBayes v. 3.2.6 [61]. Two Markov Chain Monte Carlo algorithm (MCMC) runs were conducted simultaneously with 200,000 generations and four Markov chains, starting from random trees, sampling trees every 100 generations, and discarding the fIRst 10% of samples as burn-in. The phylogenetic trees were edited and visualized using iTOL v. 3.4.3 (http://itol.embl.de/itol.cgi) (accessed on 10 December 2022).

### 4.8. Divergence Time Estimation

The dataset of 115 chloroplast genomes of Zingiberales was analyzed using GTR + G + I model determined in jModelTest v. 2.1.10 [59] to search for the best tree topology. Divergence time estimation of Zingiberales was performed by using MCMC tree in PAML v. 4.4 [62]. Three fossil records and one calibration point was obtained and used in this divergence time estimation. *Zingiberopsis attenuate* was used as a mean age of 65 Million years ago (Mya) for the crown age of family Zingiberaceae [63]. *Ensete oregonense* was applied to calibrate the crown age of *Ensete* and *Musella* with a mean age of 43 Mya [64]. *Spirematospermum chandlerae* was used to calibrate the crown age of Zingiberales with a mean age of 83.5 Mya [65]. Each fossil calibration point was assumed to follow a normal distribution with a standard deviation of 2 and an offset of 2, resulting in 63.1–70.9 Mya, 41.1–48.9 Mya, and 81.6–89.4 Mya, 95% intervals, respectively. Then, the stem root of Zingiberales was set with a mean age of 100 Mya and a standard deviation of 5 (90.2–110 Mya, 95% intervals), based on previous studies [9,66]. The ML tree constructed from chloroplast genome sequences was used as a starting tree for MCMC run. MCMC run was set 1,000,000 generations, sampling every 100 generations, and removing the fIRst 10% generations as burn in. Divergence time estimation was calculated by parameters of clock = 2 and model = 0, with 95% highest posterior density (HPD) intervals.

## 5. Conclusions

In this study, we sequenced, assembled and compared structural characteristics of 22 complete chloroplast genomes from 21 Zingiberales species, studied the molecular evolution of chloroplast genomes in Zingiberales, reconstructed the phylogenetic relationships of Zingiberales with a high-resolution backbone, and inferred the phylogenetic divergence time of Zingiberales. The newly sequenced 22 chloroplast genomes of Zingiberales had a typical quadripartite structure and contained 111–112 different genes, including 79 protein-coding genes, 28–29 tRNA genes and 4 rRNA genes, with chloroplast genome length of 161,303–163,979 bp. Comparative analyses of 111 complete chloroplast genomes of Zingiberales identified 3 highly divergent regions, which can be used as candidate markers for phylogenetic analyses and species identification. Both ML and BI phylogenetic trees based on chloroplast genome sequences identified a strongly supported clade of ((Cannaceae, Marantaceae), (Costaceae, Zingiberaceae)), sister to (Musaceae, (Heliconiaceae, (Lowiaceae, Strelitziaceae))) (MLBS = 94–100%, and BIPP = 0.93–1.0). Reconstruction divergence time of Zingiberales revealed that the stem and crown nodes of Zingiberales were approximately 100.1 Mya (95% HPD: 92.2–109.6 Mya), and 85.0 Mya (95% HPD: 81.6–89.3 Mya), respectively. The major family-level lineages of Zingiberales becoming ranged from 46.8 to 80.5 Mya. In addition, 32 genes were under positive selection at levels of amino acids with high posterior probabilities among 111 complete chloroplast genomes of Zingiberales. All the obtained genomic resources in this study will contribute to future studies of species identification, phylogeny, molecular evolution, and conservation of Zingiberales species.

## Figures and Tables

**Figure 1 ijms-24-15031-f001:**
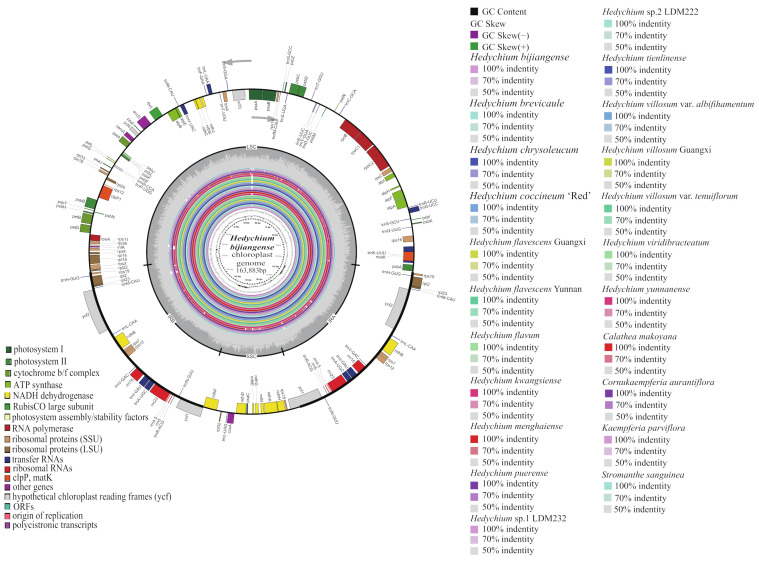
Complete chloroplast genome map of *H. bijiangense* (GenBank OP805589; the outermost three rings) and CGView comparison of 22 complete chloroplast genomes in Zingiberales (the inter rings with different colors). Genes belonging to different functional groups are shown in different colors in the outermost fIRst ring. Genes shown on the outside of the outermost fIRst ring are transcribed counter-clockwise and on the inside clockwise. Gray arrowheads indicate the direction of the genes. The tRNA genes are indicated by a one-letter code of amino acids with anticodons. The outermost second ring with darker gray corresponds to GC content, whereas the outermost third ring with lighter gray corresponds to AT content of *H. bijiangense* chloroplast genome. The innermost fIRst black ring indicates the chloroplast genome size of *H. bijiangense*. The innermost second and third rings indicate GC content and GC skew deviations in the chloroplast genome of *H. bijiangense*, respectively: GC skew + indicates G > C, and GC skew – indicates G < C. From the innermost fourth color ring to the outwards 25th ring in turn: *H. bijiangense* OP805589, *H. brevicaule* OP805581, *H. chrysoleucum* OP805577, *H. coccineum* ‘Red’ OP805574, *H. flavescens* Guangxi OP805591, *H. flavescens* Yunnan OP805575, *H. flavum* OP805588, *H. kwangsiense* OP805586, *H. menghaiense* OP805587, *H. puerense* OP805578, *H.* sp.1 LDM232 OP805583, *H.* sp.2 LDM222 OP805582, *H. tienlinense* OP805590, *H. villosum* var. *albifihamentum* OP805580, *H. villosum* Guangxi OP805584, *H. villosum* var. *tenuiflorum* OP805576, *H. viridibracteatum* OP805579, *H. yunnanense* OP805585, *C. makoyana* OP805573, *C. aurantiflora* OP805593, *K. parviflora* OP805592, *S. sanguinea* OP805594; chloroplast genome similar and highly divergent locations are represented by continuous and interrupted track lines, respectively. LSC, large single-copy region; SSC, small single-copy region; and IR, inverted repeat.

**Figure 2 ijms-24-15031-f002:**
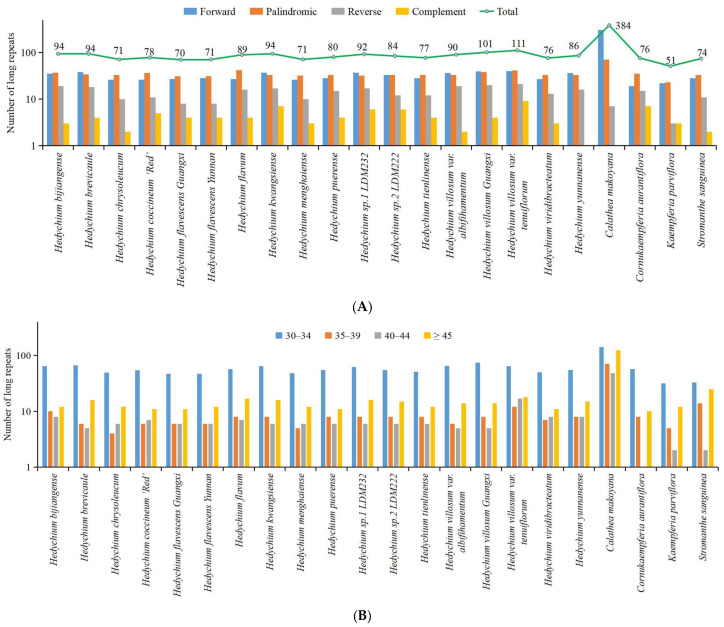
Analysis of long repeats in the 22 newly sequenced complete chloroplast genomes of Zingiberales. (**A**) Total number of four long repeat types. (**B**) Length distribution of long repeats in each sequenced chloroplast genome.

**Figure 3 ijms-24-15031-f003:**
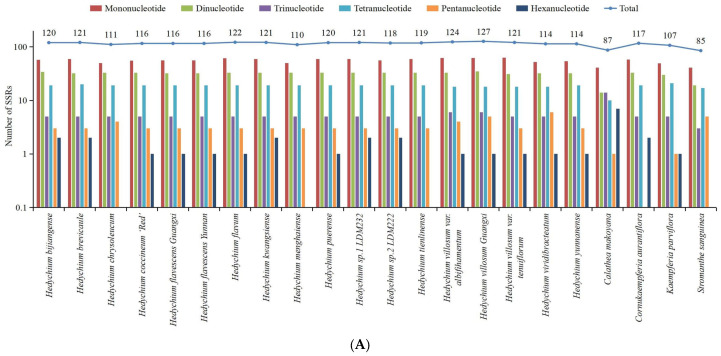
Types and distribution of SSRs in 22 newly sequenced complete chloroplast genomes of Zingiberales. (**A**) Number of different SSR types. (**B**) Number of identified SSR motifs in different repeat class types. SSR, simple sequence repeat.

**Figure 4 ijms-24-15031-f004:**
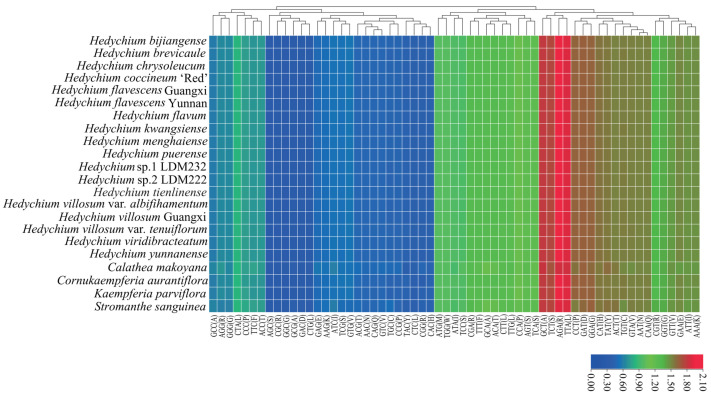
Heat map for the relative synonymous codon usage values of the 22 newly sequenced complete chloroplast genomes of Zingiberales.

**Figure 5 ijms-24-15031-f005:**
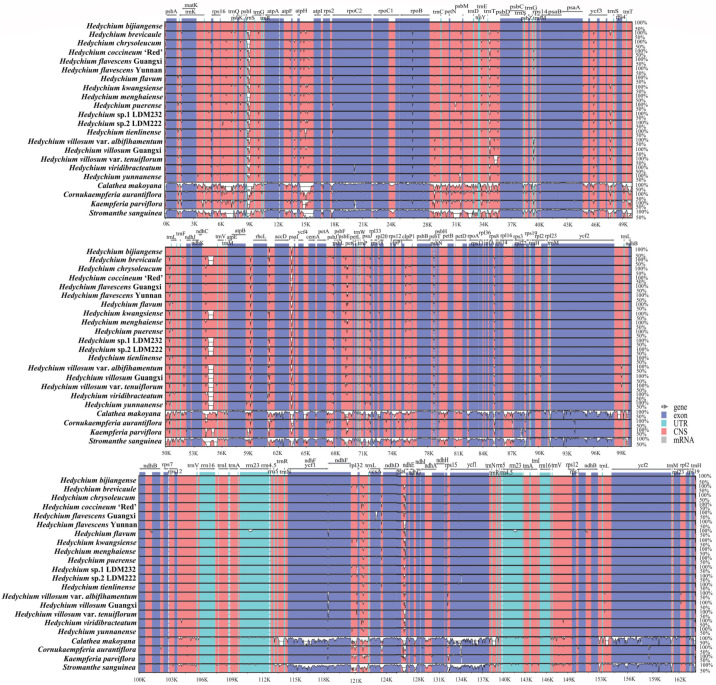
Visualization of the alignment of the 22 newly sequenced complete chloroplast genomes of Zingiberales. The chloroplast genome of *H. bijiangense* is used as the reference. The y-axis represents the percent identity ranging from 50% to 100%. The x-axis depicts sequence coordinates within the chloroplast genome. Purple bars represent exons, sky-blue bars represent untranslated regions (UTRs), red bars represent non-coding sequences (CNS), grey bars represent mRNA and white regions represent sequence differences among 22 analyzed chloroplast genomes.

**Figure 6 ijms-24-15031-f006:**
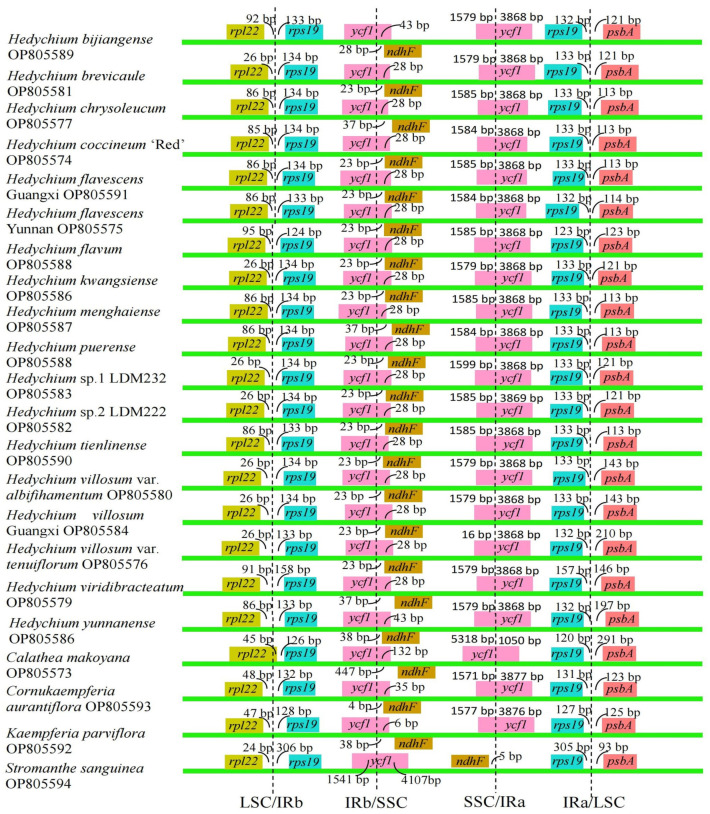
Comparison of the IR/SC borders among 22 newly sequenced complete chloroplast genomes of Zingiberales. LSC, large single-copy region; SSC, small single-copy region; IR, inverted repeat region.

**Figure 7 ijms-24-15031-f007:**
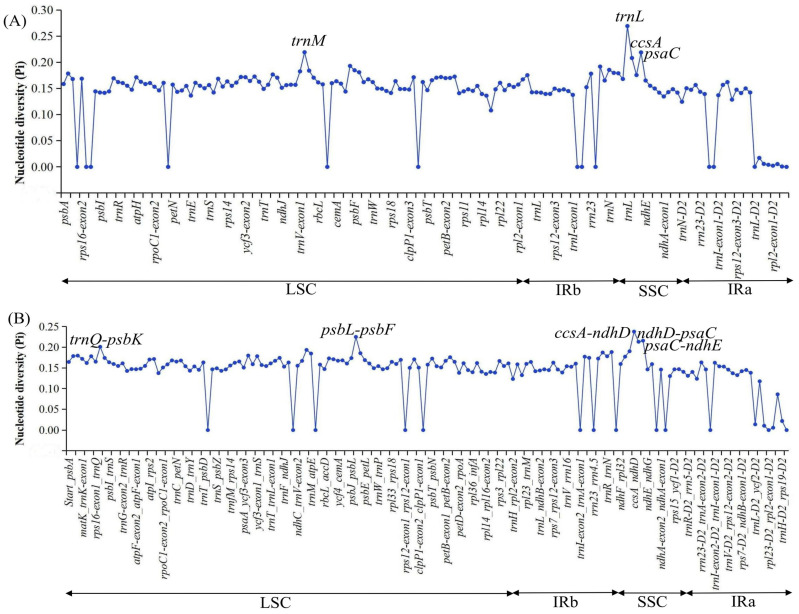
Comparison of nucleotide diversity values across the 111 complete chloroplast genomes of Zingiberales. (**A**) Protein-coding regions. (**B**) Non-coding regions.

**Figure 8 ijms-24-15031-f008:**
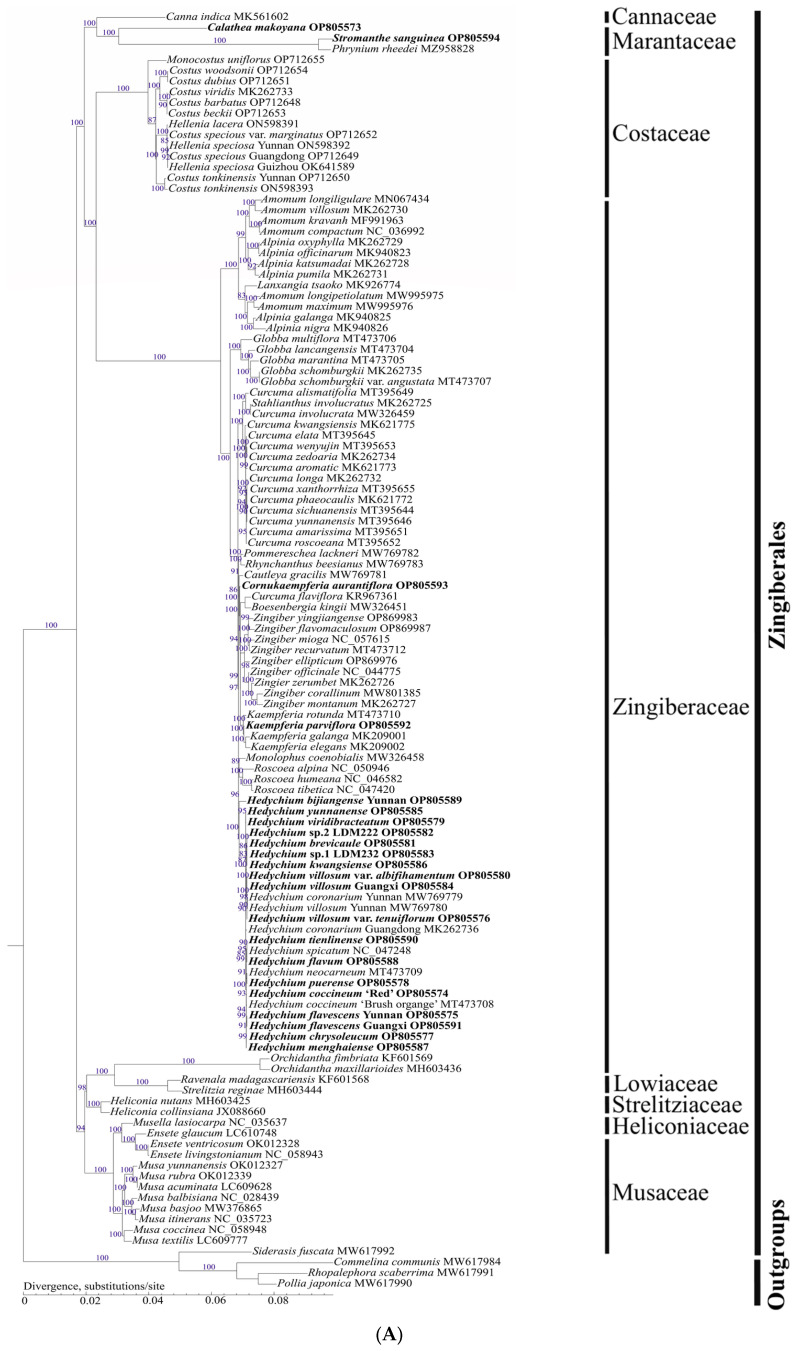
Two phylogenetic trees reconstructed from the 115 chloroplast genome sequences of Zingiberales and 4 outgroups using maximum likelihood (ML) and Bayesian inference (BI), respectively. (**A**) Phylogenetic tree reconstruction using ML method of PhyML v. 3.0. Numbers next to the branches are ML bootstrap support values. (**B**) Phylogenetic tree reconstruction using the BI method of MrBayes v. 3.2.6. Numbers next to the branches are BI probability support values. The newly sequenced 22 chloroplast genomes in this study are in bold.

**Figure 9 ijms-24-15031-f009:**
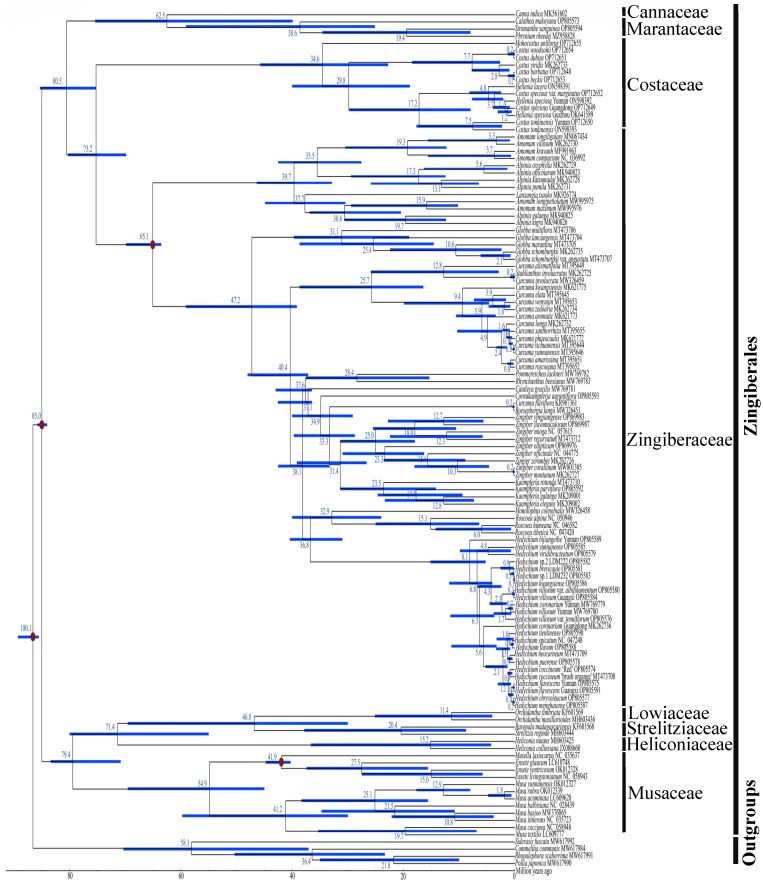
Divergence time estimation of Zingiberales based on the 115 chloroplast genome sequences. The fossil and calibration taxa are indicated with red points on the corresponding nodes. The mean divergence time of the nodes is shown at the nodes with blue. The blue bars correspond to 95% HPD of estimated divergence time, with minimum and maximum values, respectively.

**Table 1 ijms-24-15031-t001:** Basic characteristics of the newly sequenced 22 complete chloroplast genomes of Zingiberales in this study.

Species Name	GenBank Accession Number	Size (bp)	LSC (bp)	SSC (bp)	IR (bp)	GC Content (%)					Number of Genes (Different)	Number of CDS (Different)	Number of tRNA (Different)	Number of rRNA (Different)
						Total	LSC	SSC	IR	CDS				
*Hedychium bijiangense*	OP805589	163,883	88,541	15,786	29,778	36.08	33.84	29.56	41.14	36.93	133 (112)	87 (79)	38 (29)	8 (4)
*Hedychium brevicaule*	OP805581	163,438	88,016	15,862	29,780	36.09	33.86	29.52	41.14	36.91	131 (111)	87 (79)	36 (28)	8 (4)
*Hedychium chrysoleucum*	OP805577	163,977	88,603	15,816	29,779	36.08	33.84	29.53	41.15	36.93	133 (112)	87 (79)	38 (29)	8 (4)
*Hedychium coccineum* ‘Red’	OP805574	163,850	88,487	15,805	29,779	36.10	33.88	29.54	41.15	36.93	133 (112)	87 (79)	38 (29)	8 (4)
*Hedychium flavescens* Guangxi	OP805591	163,909	88,589	15,762	29,779	36.10	33.85	29.62	41.15	36.93	133 (112)	87 (79)	38 (29)	8 (4)
*Hedychium flavescens* Yunnan	OP805575	163,951	88,591	15,804	29,778	36.09	33.85	29.57	41.15	36.93	133 (112)	87 (79)	38 (29)	8 (4)
*Hedychium flavum*	OP805588	163,850	88,573	15,821	29,707	36.08	33.84	29.52	41.16	36.93	133 (112)	87 (79)	38 (29)	8 (4)
*Hedychium kwangsiense*	OP805586	163,423	88,002	15,861	29,780	36.09	33.86	29.52	41.14	36.91	133 (112)	87 (79)	38 (29)	8 (4)
*Hedychium menghaiense*	OP805587	163,979	88,603	15,818	29,779	36.08	33.84	29.53	41.15	36.93	133 (112)	87 (79)	38 (29)	8 (4)
*Hedychium puerense*	OP805578	163,941	88,561	15,822	29,779	36.08	33.84	29.52	41.14	36.92	133 (112)	87 (79)	38 (29)	8 (4)
*Hedychium* sp.1 LDM232	OP805583	163,442	88,021	15,861	29,780	36.09	33.86	29.52	41.14	36.91	133 (112)	87 (79)	38 (29)	8 (4)
*Hedychium* sp.2 LDM222	OP805582	163,319	87,916	15,843	29,780	36.11	33.88	29.52	41.14	36.91	133 (112)	87 (79)	38 (29)	8 (4)
*Hedychium tienlinense*	OP805590	163,868	88,488	15,820	29,780	36.09	33.87	29.53	41.14	36.93	133 (112)	87 (79)	38 (29)	8 (4)
*Hedychium villosum* var. *albifihamentum*	OP805580	163,442	88,059	15,807	29,788	36.11	33.86	29.57	41.16	36.92	133 (112)	87 (79)	38 (29)	8 (4)
*Hedychium villosum* Guangxi	OP805584	163,462	88,040	15,846	29,788	36.10	33.86	29.49	41.16	36.92	133 (112)	87 (79)	38 (29)	8 (4)
*Hedychium villosum* var. *tenuiflorum*	OP805576	163,359	88,139	15,678	29,771	36.12	33.86	29.71	41.14	37.11	133 (112)	87 (79)	38 (29)	8 (4)
*Hedychium viridibracteatum*	OP805579	163,338	88,032	15,746	29,780	36.12	33.89	29.61	41.15	36.93	133 (112)	87 (79)	38 (29)	8 (4)
*Hedychium yunnanense*	OP805585	163,420	88,071	15,789	29,780	36.11	33.87	29.56	41.15	36.93	133 (112)	87 (79)	38 (29)	8 (4)
*Calathea makoyana*	OP805573	166,906	91,910	20,848	27,074	36.67	34.93	30.35	42.05	37.60	134 (112)	88 (79)	38 (29)	8 (4)
*Cornukaempferia aurantiflora*	OP805593	163,305	88,197	15,640	29,734	36.17	33.92	29.90	41.15	36.93	133 (112)	87 (79)	38 (29)	8 (4)
*Kaempferia parviflora*	OP805592	163,075	87,769	15,812	29,747	36.16	33.97	29.57	41.14	36.95	133 (112)	87 (79)	38 (29)	8 (4)
*Stromanthe sanguinea*	OP805594	161,303	87,056	18,937	27,655	36.55	34.44	29.84	42.16	37.57	133 (112)	87 (79)	38 (29)	8 (4)

Note: CDS protein-coding genes, GC guanine–cytosine, LSC large single-copy region, SSC small single-copy region, and IR inverted repeat.

**Table 2 ijms-24-15031-t002:** Gene contents in the newly sequenced 22 complete chloroplast genomes of Zingiberales in this study.

Category of Genes	Group of Genes	Name of Genes
Self-replication	DNA dependent RNA polymerase	*rpoA*, *rpoB*, *rpoC1* *, *rpoC2*
	Large subunit of ribosomal proteins	*rpl2* (×2) *, *rpl14*, *rpl16* *, *rpl20*, *rpl22* (×2) ①, *rpl23* (×2), *rpl32*, *rpl33*, *rpl36*
	Small subunit of ribosomal proteins	*rps2*, *rps3*, *rps4*, *rps7* (×2), *rps8*, *rps11*, *rps12* (×2) *, *rps14*, *rps15*, *rps16* *, *rps18*, *rps19* (×2)
RNA genes	Ribosomal RNA	*rrn4.5* (×2), *rrn5* (×2), *rrn16* (×2), *rrn23* (×2)
	Transfer RNA	*trnA-UGC* (×2) *, *trnC-GCA*, *trnD-GUC*, *trnE-UUC*, *trnF-GAA*, *trnfM-CAU*, *trnG-GCC*, *trnG-UCC* *, *trnH-GUG* (×2) ②, *trnI-GAU* (×2) *, *trnK-UUU* *, *trnL-CAA* (×2), *trnL-UAA* *, *trnL-UAG*, *trnM-CAU* (×3), *trnN-GUU* (×2), *trnP-UGG*, *trnQ-UUG*, *trnR-ACG* (×2), *trnR-UCU* ③, *trnS-GCU*, *trnS-GGA*, *trnS-UGA*, *trnT-GGU, trnT-UGU*, *trnV-GAC* (×2), *trnV-UAC* *, *trnW-CCA*, *trnY-GUA*
Photosynthesis related genes	Subunits of photosystem Ⅰ	*psaA*, *psaB*, *psaC*, *psaI*, *psaJ*
	Subunits of photosystem Ⅱ	*psbA*, *psbB*, *psbC*, *psbD*, *psbE*, *psbF*, *psbH*, *psbI*, *psbJ*, *psbK*, *psbL*, *psbM*, *psbN*, *psbT*, *psbZ*, *infA*
	Subunits of cytochrome b/f complex	*petA*, *petB* *, *petD* *, *petG*, *petL*, *petN*
	Subunits of ATP synthase	*atpA*, *atpB*, *atpE*, *atpF* *, *atpH*, *atpI*
	Subunits of NADH dehydrogenase	*ndhA* *, *ndhB* (×2) *, *ndhC*, *ndhD*, *ndhE*, *ndhF*, *ndhG*, *ndhH*, *ndhI*, *ndhJ*, *ndhK*
	Subunit of rubisco	*rbcL*
Other genes	Subunit of acetyl-coA-carboxylase	*accD*
	c-type cytochrome synthesis gene	*ccsA*
	Envelop membrane protein	*cemA*
	Protease	*clpP* **
	Maturase	*matK*
Genes of unknown function	Conserved open reading frames	*ycf1* (×2), *ycf2* (×2), *ycf3* **, *ycf4*

Note: *: gene containing one intron; **: gene containing two introns; (×2): gene with two copies; (×3): gene with three copies; ①: *rpl22* has two copies only in the chloroplast genome of *Calathea makoyana*; ②: *trnH-GUG* has only one copy in the chloroplast genome of *Hedychium brevicaule*; ③: *trnR-UCU* is missing in the chloroplast genome of *H. brevicaule*.

**Table 3 ijms-24-15031-t003:** Variable site analyses of the 111 complete chloroplast genomes of Zingiberales.

Regions	Length	Variable Sites		Informative Sites		Nucleotide Diversity
		Number	%	Number	%	
LSC	88,541	65,855	74.38	54,324	61.35	0.1751
SSC	15,786	11,757	74.48	9131	57.84	0.1621
IRa	29,778	13,210	44.36	12,150	40.80	0.1455
IRb	29,778	20,298	68.16	17,141	57.56	0.1534
Complete chloroplast genome	163,883	111,120	67.80	92,746	56.59	0.1565

**Table 4 ijms-24-15031-t004:** Positively selected sites detected in the 111 complete chloroplast genomes of Zingiberales.

Gene Names	Positively Selected Sites Pr (ω > 1)
*atpA*	190 Q 0.981 *, 255 R 0.955 *, 459 V 0.998 **
*atpB*	83 M 0.975 *, 132 P 0.998 **, 143 L 0.996 **, 315 E 0.984 *
*atpF*	49 L 0.959 *, 124 F 0.986 *, 178 A 0.966 *
*atpI*	24 L 0.984 *, 65 D 0.990 *
*clpP*	19 S 1.000 **, 21 V 0.999 **, 22 E 1.000 **, 38 S 0.996 **, 56 P 1.000 **, 73 S 0.968 *, 114 A 1.000 **, 128 G 0.989 *, 133 S 1.000 **, 136 A 0.985 *, 158 Y 0.957 *, 162 Y 0.982 *, 186 G 0.986 *, 187 I 0.966 *, 189 F 0.999 **
*matK*	92 V 0.955 *, 106 R 0.959 *, 146 W 0.985 *, 279 T 0.971 *, 294 F 0.985 *, 296 R 0.997 **, 413 P 0.970 *
*ndhA*	132 F 0.982 *
*ndhB*	13 F 0.996 **, 163 T 1.000 **, 228 P 0.993 **, 241 G 1.000 **, 379 S 1.000 **
*ndhC*	98 L 1.000 **
*ndhD*	422 A 0.989 *
*ndhF*	504 G 0.998 **, 623 F 0.955 *, 630 K 0.957 *
*ndhI*	165 D 0.967 *
*petB*	1 M 1.000 **, 2 S 1.000 **
*petD*	106 T 0.990 **
*psaA*	152 S 0.968 *, 261 L 0.983 *, 292 A 0.988 *
*psaI*	4 F 0.992 **
*psbA*	155 A 0.980 *
*psbB*	494 A 0.981 *
*psbD*	3 V 1.000 **, 4 A 1.000 **, 5 L 1.000 **
*rbcL*	219 L 0.999 **, 225 I 1.000 **, 226 Y 1.000 **, 240 L 1.000 **, 255 V 1.000 **, 407 L 0.998 **, 424 L 0.999 **, 449 S 1.000 **
*rpl20*	69 N 0.987 *, 70 K 1.000 **, 77 R 0.984 *
*rpl23*	62 K 0.987 *
*rpoB*	787 G 0.953 *
*rpoC1*	141 D 0.950 *
*rpoC2*	398 L 0.995 **, 582 S 0.957 *, 622 W 0.968 *, 673 Y 1.000 **, 707 S 1.000 **, 811 T 0.952 *, 972 I 0.987 *, 1119 W 0.990 **
*rps3*	83 L 0.986 *
*rps4*	157 P 0.956 *
*rps7*	43 L 0.998 **, 51 Q 0.963 *, 81 S 1.000 **, 112 P 0.980 *, 131 S 0.994 **
*rps8*	90 R 0.996 **
*rps15*	72 R 0.994 **
*ycf3*	109 C 0.981 *, 110 H 1.000 **
*ycf4*	158 L 0.975 *, 175 R 0.976 *

Note: * and ** indicate a posterior probability higher than 0.95 and 0.99, respectively.

## Data Availability

The data presented in this study were submitted to NCBI repository (https://www.ncbi.nlm.nih.gov) (accessed on 10 November 2022), with accession numbers OP805573–OP805594. Other chloroplast genomes for phylogenetic analysis and divergence time estimation can be obtained from NCBI, and their accession numbers are in Appendix A.

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
