# Peer review of "Comparative Chloroplast Genomics of 21 Species in Zingiberales with Implications for Their Phylogenetic Relationships and Molecular Dating"

_ijms, 2023, doi:10.3390/ijms241915031_

Round 1

Reviewer 1 Report

Authors sequenced chloroplast genomes of plants representing 21 species in Zingiberales. Moreover Authors assembled and compared structural characteristics of these sequences, enabling the estimation of divergence time. Results significantly increased the available information related to chloroplast genome sequences of Zingiberales and their phylogenetic relationships. Research is original and relevant to the field. Research is well planned and performed. Results suport conclusions. References are appropriate.

Some minor comments should be addressed:

1. Correct typographical errors in lines 330, 331, 599, 600; remove the excessive brackets.

2. Section 4.1; provide the volume and concentration of DNA libraries.

3. Correct two sentences:

The word „and” should be placed before the word „showed”- the word „and” is :written in blue font.

Lines 442-445:

Compared with previous studies based on morphology, chloroplast genome genes and nuclear ITS [1,3,23], incomplete chloroplast genomes and nuclear genes [4–6,20], our results sampled more species using chloroplast genomes sequences and showed high resolution of phylogenetic relationships in Zingiberales (Figure 8).

The word „showed” or „presented” (blue font) should be used instead of word were (red font)

Lines 417-419

By contrast, our results showed/presented were less genes under positive selection than results in a previous study [34], in which most protein coding genes (62) underwent positive selection pressures among 14 Araceae species.

Minor editing of English language required.

Author Response

  1. Correct typographical errors in lines 330, 331, 599, 600; remove the excessive brackets.

Response: The brackets in lines 330-331, and 599-600 are necessary. These brackets indicated phylogenetic relationships within the two clades, including one clade of ((Cannaceae, Marantaceae), (Costaceae, Zingiberaceae)), and the other clade of (Musaceae, (Heliconiaceae, (Lowiaceae, Strelitziaceae))). Therefore, we remain the original form of these sentences.

  1. Section 4.1; provide the volume and concentration of DNA libraries.

Response: We used 1.0 μg DNA from each sample for starting DNA library construction. The volume of constructed DNA library for each sample was 25 μL and the concentration of constructed DNA library for each sample was 21.5 ng/μL. We revised section 4.1 as following:

Each qualified chloroplast DNA (1.0 μg) was used for construction a DNA library with fragments of about 350 bp, volume of 25 μL and concentration of 21.5 ng/μL, and then sequenced on an Illumina NovaSeq 6000 platform with 150 bp paired-end reads length (Biozeron, Shanghai, China).

  1. Correct two sentences:

The word „and” should be placed before the word „showed”- the word „and” is :written in blue font.

Lines 442-445:

Compared with previous studies based on morphology, chloroplast genome genes and nuclear ITS [1,3,23], incomplete chloroplast genomes and nuclear genes [4–6,20], our results sampled more species using chloroplast genomes sequences and showed high resolution of phylogenetic relationships in Zingiberales (Figure 8).

 Response: Yes, we agreed and revised.

Compared with previous studies based on morphology, chloroplast genome genes and nuclear ITS [1,3,23], incomplete chloroplast genomes and nuclear genes [4–6,20], our results sampled more species using chloroplast genomes sequences and showed high resolution of phylogenetic relationships in Zingiberales (Figure 8).

The word „showed” or „presented” (blue font) should be used instead of word were (red font)

Lines 417-419

By contrast, our results showed/presented were less genes under positive selection than results in a previous study [34], in which most protein coding genes (62) underwent positive selection pressures among 14 Araceae species.

Response: Yes, we agreed and revised as following:

By contrast, we found less genes under positive selection than results in a previous study [34], in which most protein coding genes (62) underwent positive selection pressures among 14 Araceae species.

Reviewer 2 Report

The manuscript is very well written and is evident of quality work.

Some suggestions for improvement are as follows

1. Line 18- change "showed high conservative: were highly conserved

2. Line 20- change "3 highly divergent regions comprising ccsA, psaC and psaC-ndhE, were identified" to "identified 3 highly divergent regions comprising ccsA, psaC and psaC-ndhE"

3. Line 23: change colon (:) to a full stop (.)

4. Line 75: change "has" to "is"

5. Figure 1: Correct the spelling of chloroplast (it's not "chroplast")  in the centre of the map 

6. Line 417: change "our results were" to "we found"

Some words need to be dropped off as shown in strikethrough in the attached PDF. 

Minor editing is required.

Author Response

Some suggestions for improvement are as follows

  1. Line 18- change "showed high conservative: were highly conserved

Response: Yes, we agreed and revised.

  1. Line 20- change "3 highly divergent regions comprising ccsA, psaC and psaC-ndhE, were identified" to "identified 3 highly divergent regions comprising ccsA, psaC and psaC-ndhE"

Response: Yes, we agreed and revised.

  1. Line 23: change colon (:) to a full stop (.)

Response: Yes, we agreed and revised.

  1. Line 75: change "has" to "is"

Response: Yes, we agreed and revised.

  1. Figure 1: Correct the spelling of chloroplast (it's not "chroplast")  in the centre of the map 

Response: Thanks very much for this idea. We used the correct spelling of “chloroplast” in the centre map of Figure 1.

  1. Line 417: change "our results were" to "we found"

Response: Yes, we agreed and revised.

Some words need to be dropped off as shown in strike through in the attached PDF. 

Response: Yes, we agreed and revised.

Reviewer 3 Report

The manuscript “Comparative Chloroplast Genomics of 21 Species in Zingiberales With Implications for Its Phylogenetic Relationships and Molecular Dating” described the study of sequenced chloroplast genome of 21 species among the families Zingiberaceae and Marantaceae for structural and genomic content characteristic analysis. Phylogenetics tree relationship and molecular evolution of Zingiberales was also revealed in this study. This provides important genomic resources for future study.

Major concerns

  1. Page 20, Line 488, For the total genomic DNA extraction, will the nuclear DNA together with chloroplast DNA be precipitated during the DNA extraction process, will this affect the process of chloroplast genome assembly? Did the authors find any of the chloroplast genes matching with the nucleus genome?

  1. Based on the chloroplast genome analysis, will the result of data analysis focused on the nuclear genome from Zingiberales be similar in this study?

Author Response

Major concerns

  1. Page 20, Line 488, For the total genomic DNA extraction, will the nuclear DNA together with chloroplast DNA be precipitated during the DNA extraction process, will this affect the process of chloroplast genome assembly?

Response: Page 20. line 488, the modified sucrose gradient centrifugation method was used for the extraction of total chloroplast genomic DNA. The chloroplast DNA abundance in sequencing reads was very high. The presence of small amounts of nuclear DNA does not affect chloroplast genome assembly, because reads coverage depth varies widely, with chloroplast genome coverage very high (hundreds of thousands of layers), whereas nuclear genome coverage depth relatively low (generally less than 10 ×).

Did the authors find any of the chloroplast genes matching with the nucleus genome?

Response: The chloroplast genomes communicate with the mitochondrial and nuclear genomes in small amounts, mainly in the intergenic regions. Because of the different boundaries and the different depth of coverage, reads can assemble across the boundaries. There is no situation that the assembly of communication segments is incomplete.

  1. Based on the chloroplast genome analysis, will the result of data analysis focused on the nuclear genome from Zingiberales be similar in this study?

Response: We did not do the nuclear genome analysis from Zingiberales. However, a previous study (Timilsena et al. 2022) reported phylogenomic relationships of monocot including Zingiberales using 602 single-copy nuclear genes and 1375 BUSCO genes. This report found that Musaceae was sister to the clade containing Heliconiaceae, which was in turn sister to Strelitziaceae and Lowiaceae with weakly to strongly supports (bootstrap = 69–100%, and posterior probabilities = 0.6-1.0). This situation may need more samples of Zingiberales, and more nuclear genes/data to improve the resolutions.

In this study, we used more samples of Zingiberales based on chloroplast genomes analyses. The results indicated a strongly supported backbone of phylogenetic relationships of Zingiberales. Cannaceae was sister to Marantaceae, forming a clade that was collectively sister to the clade of (Costaceae, Zingiberaceae) with strong support (BS = 100%, and PP = 0.99–1.0); Heliconiaceae was sister to the clade of (Lowiaceae, Strelitziaceae), then collectively sister to Musaceae with strong support (BS = 94–100%, and PP = 0.93–1.0); the clade of ((Cannaceae, Marantaceae), (Costaceae, Zingiberaceae)) was sister to the clade of (Musaceae, (Heliconiaceae, (Lowiaceae, Strelitziaceae))) with robust support (BS = 100%, and PP = 1.0). The result of current study was in agreement with the backbone of the previous study (Timilsena et al. 2022), but we improved the resolutions of phylogenomic relationships of Zingiberales.

Timilsena, P.R.; Wafula, E.K.; Barrett, C.F.; Ayyampalayam, S.; McNeal, J.R.; Rentsch, J.D.; McKain, M.R.; Heyduk, K.; Harkess, A.; Villegente, M.; et al. Phylogenomic resolution of order- and family-level monocot relationships using 602 single-copy nuclear genes and 1375 BUSCO genes. Front. Plant Sci. 2022, 13, 876779.
